# Induction of Multiple Alternative Mitogenic Signaling Pathways Accompanies the Emergence of Drug-Tolerant Cancer Cells

**DOI:** 10.3390/cancers16051001

**Published:** 2024-02-29

**Authors:** Frank V. Celeste, Scott Powers

**Affiliations:** 1Graduate Program in Genetics, Stony Brook University, Stony Brook, NY 11794, USA; 2Department of Pathology, Stony Brook University, Stony Brook, NY 11794, USA

**Keywords:** drug-tolerant cells, persisters, acquired resistance, targeted therapies

## Abstract

**Simple Summary:**

Targeted therapies are typically well tolerated and frequently lead to rapid tumor regression. However, the re-emergence of a drug-resistant tumors poses a significant challenge to their long-term effectiveness. A small subset of drug-tolerant “persister” cells have been identified as crucial precursors to full-fledged resistance. While various studies have pinpointed the activation of alternative signaling pathways as a key mechanism for evading therapy effects, targeting these secondary pathways alone has not yet reached the clinic. Currently, there is a pressing need for a more comprehensive profiling of the early drug-tolerant cellular landscape. This profiling is essential for understanding the underlying mechanisms driving resistance emergence and for devising more impactful therapeutic strategies.

**Abstract:**

Drug resistance can evolve from a subpopulation of cancer cells that initially survive drug treatment and then gradually form a pool of drug-tolerant cells. Several studies have pinpointed the activation of a specific bypass pathway that appears to provide the critical therapeutic target for preventing drug tolerance. Here, we take a systems-biology approach, using proteomics and genomics to examine the development of drug tolerance to EGFR inhibitors in EGFR-mutant lung adenocarcinoma cells and BRAF inhibitors in BRAF-mutant melanoma cells. We found that there are numerous alternative mitogenic pathways that become activated in both cases, including YAP, STAT3, IGFR1, and phospholipase C (PLC)/protein kinase C (PKC) pathways. Our results suggest that an effective therapeutic strategy to prevent drug tolerance will need to take multiple alternative mitogenic pathways into account rather than focusing on one specific pathway.

## 1. Introduction

The first study that demonstrated that a subpopulation of cancer cells which survive drug treatment form viable drug-tolerant cells was the study of Sharma et al., who utilized targeted inhibition of the EGFR receptor in the EGFR-mutant non-small-cell lung cancer (NSCLC) line PC9 [1]. Drug-tolerant cells, which are also referred to as drug-tolerant persisters, have also been found for other targeted therapeutic treatments such as those for BRAF-mutant melanoma [2], androgen-receptor-driven prostate cancer [3], and HER2-amplified breast cancer [4], as well as chemotherapy [5]. Several lines of evidence, including the reversibility of drug tolerance upon the withdrawal of the drug, suggest that epigenetic changes underlie the development of drug tolerance. Indeed, a role for the KDM5A histone demethylase in mediating drug tolerance in PC9 cells was observed in the Sharma et al. study [1], and similarly, KDM5B has been shown to mediate the development of BRAF inhibitor tolerance in melanoma [6]. Many other studies of the epigenetic changes in the development of drug-tolerant cells have been recently reviewed [7].

Although it is often assumed that the vast majority of oncogene-addicted cancer cells undergo apoptosis upon treatment with targeted therapeutics, fewer than 50% of PC9 cells do so in response to gefitinib, and less than 20% of the melanoma cell line SKMEL28 undergoes apoptosis in response to dabrafenib [4,8]. The remaining surviving cells are mostly quiescent. Time-lapse microscopy studies have demonstrated that a subpopulation of drug-tolerant lung cancer and melanoma cells are not completely quiescent and continue to cycle during drug treatment, albeit at a much slower rate [9,10]. Similar conclusions have been reached by two recent studies that have used fluorescent biosensors to track the fate of individual cells during the development of drug tolerance. In one study, a biosensor that measured *CDK2* activity was used to track individual BRAF-mutant melanoma cells as they developed into drug-tolerant cells in response to the BRAF inhibitor dabrafenib [11]. Following four days of drug treatment, they found, via time-lapse microscopy, that most cells remained in a quiescent CDK2low state, whereas other cells initially entered a CDK2low state but, over time, slowly built up CDK2 activity and underwent cell division [11]. Another group utilized time-lapse microscopy of PC9 cells treated with an EGF inhibitor for 14 days and found that >90% of the surviving cells never divided, and that only a small subset (approximately 0.5%) were able to form colonies of six or more cells [12]. These results were confirmed via flow cytometric analysis of the cells’ proliferation history through the dilution of a doxycycline-induced histone 2B-mCherry transgene [12]. Not all drug-tolerant cells are proliferative; most HER2-amplified breast cancer cell lines that form drug-tolerant cells following HER2 inhibition do not proliferate at all for at least 8 weeks but can resume proliferation if HER2 inhibition is withdrawn [4]. 

Several groups have found alternative signaling mechanisms by which drug-tolerant cancer cells can survive and even bypass the mitogenic block generated by targeted therapeutics. For example, the inhibition of mutant-EGFR in lung adenocarcinoma cells or mutant-BRAF in thyroid cancer cells was shown to induce the secretion of IL6 and the subsequent activation of STAT3, and in both cases, blocking the STAT3 function largely prevented the emergence of drug-tolerant cells [13,14]. Inhibition of mutant EGFR in lung adenocarcinoma cells has been shown to cause increased expression of FGFR3, and inhibition of FGFR3 was shown to block the formation of drug-tolerant cells [15]. Additionally, inhibition of mutant EGFR with the third-generation inhibitor osimertinib in lung adenocarcinoma cells has been shown by one group to cause the up-regulation of the tyrosine kinase receptor AXL [16], and by another group to cause the up-regulation of the mitotic protein kinase AURKA [17]. In both cases, the studies found that the inhibition of their target of interest prevented the emergence of drug-tolerant cells.

One of the primary motivations for these studies has been to discover a therapeutic strategy that could be used to block the development of drug-tolerant cells. However, disruption of a single bypass survival and/or mitogenic signaling pathway may not fully eliminate drug-tolerant cells if there is an activation of other alternate growth signaling pathways that remain intact. In this study, we set out to determine the entire scope of the mitogenic signaling pathways that are up-regulated in response to the blockade of mutant EGFR and mutant BRAF signaling.

## 2. Results

### 2.1. Development of Features of Drug Tolerance in PC9 and SKMEL28 Cells

Some of the features of drug-tolerant persisters that have been observed repeatedly include cell cycle accumulation in G1, induction of autophagy and senescence, and reversibility following drug withdrawal [7]. In order to validate the development of drug tolerance phenotypes in our studies, we examined *EGFR*-mutant PC9 NSCLC cells treated with the EGFR inhibitor gefitinib, exactly as described in the study by Sharma et al. [1]. We observed, as they did, the accumulation of cells in G1 after several days in gefitinib (Appendix A). Most importantly, after an extended period (50 days), we were able to detect something in the order of 500 small colonies in the presence of gefitinib out of a total of 100,000 plated cells, in close agreement with previous estimates of the persister formation of PC9 cells (Appendix A) [1,12]. We also found that PC9 cells treated with gefitinib, as well as BRAF-mutant SKMEL28 cells treated with dabrafenib, acquired senescent-like properties. Additionally, we found that the presence of senescence-associated b-galactosidase in dabrafenib-treated SKMEL28 was reversible once cells were detached and replated in media without drugs (Appendix A). 

### 2.2. Numerous Transcriptional Alterations Are Shared between EGFR-Mutant Lung Cancer Cells during EGFR Inhibition and BRAF-Mutant Melanoma Cells during BRAF Inhibition

Transcriptional changes are critical to a cancer cell’s ability to tolerate targeted therapy; however, it is unclear if the changes are common between different cancer types and inhibitors. We performed RNA-seq on EGFR-mutant PC9 NSCLC cells treated with the EGFR inhibitor gefitinib and BRAF-mutant SKMEL28 melanoma cells treated with the BRAF inhibitor dabrafenib to investigate the scope of the transcriptional changes in each system and to uncover shared mechanisms of drug tolerance. Differential expression analysis revealed 2179 differentially expressed genes (DEGs) (1524 up-regulated and 655 down-regulated) in PC9 cells treated with gefitinib for 3 days and 2219 DEGs (1674 up-regulated and 545 down-regulated) in PC9 cells treated with gefitinib for 9 days compared to untreated controls. In SKMEL28 cells cultured in dabrafenib, 4926 genes (2431 up-regulated and 2495 down-regulated) were differentially expressed at day 3 and 4581 (2463 up-regulated and 2118 down-regulated) genes were differentially expressed at day 9 compared to untreated controls. Additionally, we found that 24 hallmark gene sets were significantly enriched in 9-day gefitinib-treated PC9 cells and that 13 hallmark gene sets were significantly enriched in 9-day dabrafenib-treated SKMEL28 cells compared to untreated cells when using a strict cut-off of FDR < 0.05. Interestingly, 11 hallmark gene sets were significantly enriched in both drug-treated cell lines.

We performed hierarchical clustering of the 544 genes (422 up-regulated and 122 down-regulated) that were identified as commonly altered in both gefitinib-treated PC9 cells and dabrafenib-treated SLMEL28 cells (Figure 1a,b). Narrowing our focus to genes that have a log2-fold change of at least 2.0, we found alterations in 150 genes that are shared between cell lines (Figure 1c). Unique changes occur in each cell line, with 93 genes significantly altered in PC9 cells treated with gefitinib and 523 genes significantly altered in SKMEL28 cells treated with dabrafenib. A very small number of genes showed significant changes in opposite directions between cell lines. Eighteen genes showed significant up-regulation in PC9 cells and significant down-regulation in SKMEL28 cells, whereas five genes showed significant up-regulation in SKMEL28 cells and significant down-regulation in PC9 cells (Figure 1c). 

Principal component analysis (PCA) revealed the distinct separation of gefitinib-treated and untreated PC9 cells by the first principal component, which explained over 74% of the variance (Figure 1d). Interestingly, when we performed PCA on SKMEL28 cells in dabrafenib, the first principal component did not completely separate samples according to treatment status alone (Figure 1e). These findings suggest that many transcriptional changes are shared between different cancer types adapting to various targeted therapies, although the exact timing of these changes may be unique to each cell line and drug combination.

### 2.3. Proteomic and Phosphoproteomic Alterations Accompany the Development of Drug Tolerance

We next sought to investigate proteomic and phosphoproteomic changes involved in the development of drug tolerance by performing reverse phase protein array (RPPA) analysis on PC9 cells cultured in gefitinib for 0, 3, 6, and 9 days. Of the 471 antibodies included in the panel, we saw significant changes in 305 protein and/or phosphoprotein levels at one or more time points during gefitinib treatment. We performed unbiased hierarchical clustering to visualize changes in the 55 phosphoproteins that we identified as significantly altered in gefitinib-treated cells (Figure 2a). We integrated our transcriptomic and proteomic datasets by comparing the log2-fold change of mRNA transcripts to differences in protein levels, each in 9-day gefitinib-treated versus untreated PC9 cells. We observed a positive correlation of the 336 genes/proteins featured in both datasets (Figure 2b). The labeled points show both a significant increase in protein level from RPPA analysis and a log-fold change of >2 in our RNA-seq dataset. Fibronectin was selected from these points to illustrate the large increases in both protein (Figure 2c) and transcript abundance (Figure 2d). These significant changes observed in numerous protein and phosphoprotein levels in gefitinib-treated PC9 cells complement prior transcriptomic studies and reveal potential mechanisms by which cancer cells continue to cycle in targeted therapy.

### 2.4. Targeted Inhibition of Mutationally Activated Oncogenes Induces Sustained mTOR-Pathway Suppression

We began investigating the shared effects of inhibiting mutationally activated oncogenes in both EGFR-mutant lung cancer cells and BRAF-mutant melanoma cells by analyzing changes in gene expression at 3, 6, and 9 days following addition of the EGFR inhibitor gefitinib or the BRAF inhibitor dabrafenib compared to untreated controls. Using gene set enrichment analysis (GSEA), we found that mTOR-related gene sets were more enriched in untreated controls relative to gefitinib and dabrafenib-treated cells (Figure 3a,b). Transcript levels of indicated mTOR-related genes, such as *RPS6*, *EIF4G1*, and *EEF2K*, showed significant decreases in PC9 cells in gefitinib and SKMEL28 cells in dabrafenib at multiple time points (Figure 3c,d). 

We then examined the individual protein and phosphoprotein levels of mTOR pathway members. RPPA analysis revealed that total protein levels of mTOR did not significantly change, but phosphorylated mTOR levels decreased during gefitinib treatment of PC9 cells (Figure 3e). Downstream targets of mTOR, including eIF4E, p70-S6K, and S6, showed significant decreases in the amount of phosphorylated protein and less so of total protein (note the difference in scales) (Figure 3e). When immunoblotting was performed on SKMEL28 cells in dabrafenib to complement protein array results obtained in PC9 cells treated with gefitinib, we observed a significant decrease in phospho-S6 within 24 h of dabrafenib treatment that was sustained for 3 days (Figure 3f) without a corresponding decrease in total S6 protein (Figure 3g). Taken together, these results indicate that treating susceptible cells with corresponding targeted therapies results in a sudden, sustained decrease in mTOR signaling.

### 2.5. Targeted Inhibition of Mutationally Activated Oncogenes Induces IGF-Pathway Activation

To better understand how cells continue to proliferate despite a crash in mTOR signaling, we began to search for alternate mitogenic growth pathways that were up-regulated in drug-treated cells. GSEA revealed that genes associated with the insulin-like growth factor (IGF) signaling cascade are significantly up-regulated in PC9 cells treated with gefitinib (Figure 4a) and SKMEL28 cells treated with dabrafenib for 9 days relative to untreated controls (Figure 4b). In gefitinib- (Figure 4d) or dabrafenib-treated cells (Figure 4e), we observed significant increases in the individual transcript levels of various IGF-related genes such as *IGFBP3*, *IGFBP5*, *IGFBP7*, and most notably *IGF1R*. Total protein levels of IGFRB did not appear to significantly change during the 9-day gefitinib exposure of PC9 cells (Figure 4c). However, we observed significant increases in the amount of phosphorylated IGF1R (Y1135/6) in PC9 cells treated with gefitinib for 9 days (Figure 4c). We also observed significant increases in IGFBP3 at days 3 and 6 of gefitinib treatment (Figure 4c), coinciding with the increase observed in the earlier *IGFBP3* transcript levels. These results, obtained in multiple cancer contexts, expand on the previous work implicating IGF1R in anti-EGFR drug tolerance. 

### 2.6. Targeted Inhibition of Mutationally Activated Oncogenes Induces PLC/PKC Signaling

We observed increased phosphoinositide-specific phospholipase C (PLC) activity in drug-treated cells. PLC is often a key player in transmembrane signaling. GSEA performed using genes associated with GPCR signaling revealed strong enhancement in gefitinib-treated PC9 cells relative to untreated controls (Figure 5a). Further GSEA studies performed on SKMEL28 cells showed a strong enrichment of genes associated with the regulation of phospholipase activity in the drug-treated group (Figure 5b). The transcript levels of several genes associated with increased PLC/PKC signaling, such as *ADCY6*, *GNAL*, *GNAS*, *PDE1C*, and *PLCG1*, showed significant increases in PC9 cells in gefitinib (Figure 5d) and SKMEL28 cells in dabrafenib (Figure 5e) at multiple time points. Phosphorylation of PLCG2 at tyrosine 759 is a marker of the active state, and increased levels were observed in PC9 cells at days 6 and 9 of gefitinib treatment (Figure 5c). Additionally, we observed increase in PKCA and phospho-PKCA/B (T638/6431) in gefitinib-treated PC9 cells. Together, these findings indicate activation of the PKC/PLC signaling pathway upon gefitinib or dabrafenib treatment, which, to our knowledge, is the first time this has been documented.

### 2.7. Targeted Inhibition of Mutationally Activated Oncogenes Induces STAT3 Activation and YAP Activation

We also saw the activation of two pathways previously observed by others. Our GSEA studies revealed that STAT3 signaling was significantly enriched in PC9 treated with gefitinib (Appendix A) or SKMEL28 cells treated with dabrafenib (Appendix A) for 9 days when compared to untreated controls. Transcript levels of STAT3-associated genes, including *JAK1* and *BCL6*, were also significantly increased in both gefitinib-treated PC9 cells (Appendix A) and dabrafenib-treated SKMEL28 cells (Appendix A). At days 3 and 9 of gefitinib treatment, total protein levels of STAT3 did not differ significantly from untreated PC9 cells (Appendix A). STAT3 is considered active when phosphorylated at tyrosine 705, which induces dimerization and nuclear translocation. Interestingly, we observed a significant increase in the amount of phosphorylated STAT3 (Y705) following 9 days of gefitinib treatment (Appendix A). In SKMEL28 cells, we also observed an increase in the amount of phosphorylated STAT3 (Y705) within 3 days (Appendix A), while the amount of total STAT3 protein remained steady (Appendix A). From these results, we conclude that increased STAT3 signaling accompanies the emergence of drug tolerance in various contexts. 

We also observed activation of YAP signaling. GSEA studies show a strong enrichment of genes associated with YAP transcriptional signature in PC9 cells treated with gefitinib (Appendix A) or SKMEL28 treated with dabrafenib (Appendix A), when compared to untreated controls. Total YAP protein levels do not significantly change in PC9 cells during 9-day treatment with dabrafenib (Appendix A). YAP phosphorylated at serine 127 is sequestered in the cytoplasm. Interestingly, we observed a significant decrease in the level of phospho-YAP (S127) at each time point during gefitinib treatment (Appendix A). Indeed, transcript levels of numerous YAP-related genes are significantly higher in drug-treated cells compared to untreated controls of both PC9 (Appendix A) and SKMEL28 (Appendix A). Proteomic data also revealed increased protein levels of PAI-1 (Appendix A), a common YAP-associated protein, in PC9 cells treated with gefitinib for 3, 6, and 9 days. These results suggest the YAP signaling is induced upon targeted blockade in oncogene-dependent cells. 

### 2.8. Single-Cell RNA-Seq Reveals Simultaneous Up-Regulation of Markers of Multiple Alternate Mitogenic Signaling Pathways

To determine whether the activation of the multiple alternate mitogenic signaling pathways we observed in bulk experiments (i.e., RNA-seq, RPPA) occurred simultaneously in individual cells or occurred in separate sub-populations, we performed single-cell RNA seq on PC9 cells in gefitinib and SKMEL28 cells in dabrafenib for 0, 24, 48, or 72 h. Gene–gene plots showed that select marker genes for IGF and STAT3 signaling (Appendix A), IGF and YAP signaling (Appendix A), IGF and PKC/PLC signaling (Appendix A) were simultaneously up-regulated in PC9 cells in gefitinib for 72 h compared to untreated controls. Additionally, we observed simultaneous up-regulation of the same pathways in SKMEL28 cells in dabrafenib for 72 h compared to untreated SKMEL28 cells (Appendix A). The simultaneous up-regulation of multiple marker genes indicates that the enriched pathways are simultaneously up-regulated in single cells rather than being up-regulated in different sub-populations prior to pooled analysis. 

### 2.9. Autophagy Accompanies the Development of Drug Tolerance and Can Be Disrupted with Hydroxychloroquine to Reduce the Number of Drug-Tolerant Cells

To investigate the potential survival mechanisms utilized by cells in targeted therapy, we investigate the role of autophagy during the development of drug tolerance. We observed enrichment of the GO gene set “Selective Autophagy” in PC9 cells treated with gefitinib for 9 days compared to untreated controls (Figure 6a). Increased transcript levels of the common autophagy marker *ULK1* were observed in gefitinib-treated PC9 cells (Figure 6b). We found that a common protein marker used to assess autophagy, LC3A/B, was increased in gefitinib-treated PC9 cells when measured by RPPA (Figure 6c). Additionally, scRNA-seq analysis of *ULK1* and a second marker, *GABARAPL1*, also showed up-regulation in gefitinib-treated PC9 cells (Figure 6d). Direct visualization of autophagic flux was performed with the CYTO-ID live cell autophagy detection kit. Increased fluorescent signal, corresponding to the increase in autophagosomes critical for autophagy, was observed in gefitinib-treated PC9 cells (Figure 6e). PC9 cells in regular media showed little to no evidence of autophagy. 

To determine if disrupting autophagy leads to fewer drug-tolerant cells, we supplemented gefitinib treatment with hydroxychloroquine. This anti-malarial drug is widely available and has been previously used to disrupt autophagy by lowering lysosomal pH, thereby blocking the key step of lysosome–autophagosome fusion [18]. Hydroxychloroquine does not exert any growth effects on PC9 cells in regular media (Figure 7a) but significantly reduces the number of surviving PC9 cells when combined with gefitinib (Figure 7b). From these results, it seems autophagy is induced upon targeted therapy exposure, and using hydroxychloroquine co-treatment is an effective way to reduce the number of drug-tolerant cells. 

### 2.10. Disruption of ATG5 Results in Decreased Autophagic Flux but Does Not Enhance Gefitinib Killing of PC9 Cells

To validate the utility of blocking autophagy in cells developing drug tolerance, we disrupted the critical ATG5–ATG12 interaction via CRISPR-induced mutations in exon 2 of ATG5 [19]. In unmodified PC9 cells, immunoblotting revealed the 55 kDa ATG5-ATG12 complex, while the CRISPR-modified clones we generated appeared to only have free ATG5 (Figure 7c). The ATG5–ATG12 complex is required for autophagy, and indeed, these clones showed a decreased ability to undergo autophagy compared to parental PC9 cells (Figure 7d). Similar to the results observed with chloroquine co-treatment, CRISPR-ATG5 cells did not show any growth differences when compared to control PC9 cells in regular media (Figure 7e). Interestingly, contrary to the results observed with chloroquine co-treatment, CRISPR-ATG5 cells survived gefitinib treatment at a higher rate than control PC9 cells (Figure 7f). Although autophagy appears to be induced by the presence of targeted therapy, and chloroquine co-treatment leads to fewer drug-tolerant cells, genetic studies cast doubt on autophagy’s importance in the development of drug tolerance.

### 2.11. Pharmacologic but Not Genetic Inhibition of BCL-XL Results in Increased Gefitinib Efficacy

Cancer cells surviving challenging environments have been shown to display features of cellular senescence, a process which overlaps with autophagy. PC9 cells were subjected to X-gal staining, a common method used to assess senescence-associated beta-galactosidase (SA-β-gal) activity, a feature unique to senescent cells. We observed the characteristic blue–green colorimetric change resulting from the SA-β-gal-mediated cleavage of X-gal in PC9 cells treated with gefitinib and SKMEL28 cells treated with dabrafenib (Appendix A). GSEA revealed an enrichment of a senescence-associated gene set in PC9 cells treated with gefitinib compared to untreated controls (Appendix A). Increases in normalized transcript abundance levels of several senescence-associated genes, such as *CDKN1A* (Appendix A) and *CDKN2B* (Appendix A), are observed in PC9 cells treated with gefitinib for 3 and 9 days compared to untreated PC9 cells. scRNA-seq analysis of *CDKN2A* and *CDKN1B* also showed up-regulation in gefitinib-treated cells PC9 cells (Appendix A). 

We co-treated PC9 cells in gefitinib with multiple “senolytic” drugs to determine if selectively targeting cells displaying features of cellular senescence would reduce the number of drug-tolerant cells. The first compound, navitoclax (ABT-263), is a BCL-2 family protein inhibitor that has been shown to selectively eliminate senescent cells [20]. Navitoclax did not appear to alter the number of PC9 cells when cultured in regular media (Appendix A). However, when added to gefitinib media, navitoclax appeared to significantly reduce the number of surviving cells (Appendix A). Rather than broadly targeting BCL-2 family proteins, A1331852 is a purported BCL-XL-specific inhibitor that showed no significant impact on PC9 growth in regular media (Appendix A). However, similar to navitoclax, co-treatment with A1331852 significantly reduced the number of PC9 cells surviving gefitinib (Appendix A). From these results, we can conclude that inhibitors targeting BCL-2 family proteins, specifically BCL-XL, significantly reduce the number of drug-tolerant cells. 

To validate the utility of disrupting BCL-XL in cells developing drug tolerance, we used CRISPR/cas9 to create a BCL-XL knockout PC9 cell line (Appendix A). There were slightly fewer CRISPR-BCL-XL cells compared to unmodified PC9 cells when cultured in regular growth media for 3 days (Appendix A). However, contrary to the results observed with navitoclax and A1331852 co-treatments, CRISPR-BCL-XL cells exhibited no differences in survival during gefitinib treatment compared to unmodified PC9 cells (Appendix A). Although senescence appears to be induced by the presence of targeted therapy, and co-treatment with senolytic compounds lead to fewer drug-tolerant cells, genetic studies cast doubt on the exact mechanism by which drug tolerance is blocked. 

### 2.12. Drug-Tolerant Cells Show Significant Gene Expression Changes Affecting AXL/GAS6, Enzymes That Utilize Glutathione as a Co-Factor, and Low-Fidelity Polymerases

Drawing from reports indicating that the up-regulation of AXL kinase and its ligand GAS6 can confer resistance to EGFR-targeted therapy in lung cancer [21,22], we investigated the expression levels of these genes in both lung and melanoma persister cells. Additionally, as glutathione serves as an essential co-factor for the lipid hydroperoxidase GPX4, crucial for persister cell survival [23], we examined the expression levels of GPX4 and various glutathione S-transferase (GST) genes, known for their role in reducing oxidative stress. Given the observed high levels of reactive oxygen species (ROS) in persister cells, we further explored the expression patterns of these genes. Moreover, given the observed elevation of repair enzymes and error-prone DNA polymerases in persister cells [24,25,26,27], we included an examination of these enzymes in our analysis. Notably, among the scrutinized genes, GAS6, GSTA4, and GSTM3 exhibited significant up-regulation in both lung and melanoma persister cells (refer to Table 1). Both lung and melanoma persisters showcased numerous additional significant alterations—fourteen and eighteen, respectively—that were not mutually shared (Table 1). 

## 3. Discussion

The ability of cancer cells to tolerate lethal doses of targeted inhibitors and form resistant tumors is a widely recognized hinderance to the long-term treatment of many cancer types. Currently, we do not fully understand how oncogene-addicted cells continue to proliferate in the presence of drugs designed to inhibit their oncogenic signaling dependencies. In this study, we sought to identify growth signaling pathways that were activated in drug-tolerant cells. 

Through a combination of transcriptomic and proteomic studies, we observed increased signaling via a phospholipase C (PLC)/protein kinase C (PKC) signaling pathway. There are several isoforms of PKC, and the phospho-specific antibody used in our study is only able to detect PKCα and PKCb, both of which are regulated by diacylglyercol produced by phospholipase C. Activation of PKCα and PKCα isoenzymes have often been linked to more malignant phenotypes, while PKCδ is thought to mediate anti-cancer effects such as apoptosis [28]. To our knowledge, we are the first to propose that activation of PKC signaling may contribute to the drug-tolerant phenotype observed in slow-growing cells adapting to targeted TKIs. 

In addition to PLC/PKC, we revealed activation of several alternate mitogenic signaling pathways, including IGF1R, STAT3, and YAP, that appear to accompany the emergence of drug tolerance (Figure 8). Interestingly, upon review of the literature, we found individual examples of each pathway’s activation. IGFBP3-mediated IGF1R activation was first observed in erlotinib-treated PC9 cells [1], and a similar mechanism was also described in BRAF (V600E) melanoma cells adapting to targeted BRAF inhibition [29]. Separate investigations of EGFR inhibition in NSCLC [14] and ovarian cancer [30] models identified activation of STAT3 as being critical to the drug-tolerant state. More recently, YAP activation was observed in drug-tolerant PC9 cells co-treated with EGFR and MEK inhibitors [31] and several other cell line/anti-EGFR therapy combinations [32]. Despite many of these studies being conducted in highly similar contexts, each concludes by promoting the pharmacological inhibition benefits of their individual proposed mechanism of drug tolerance. Our study is the first to observe the activation of several diverse pathways in cancer cells adapting to targeted EGFR or BRAF inhibition. 

Our observation that drug-tolerant cells activate multiple alternate growth signaling pathways raises an interesting question: are these pathways simultaneously up-regulated in each surviving cell, or are sub-populations of cells each strongly up-regulating a single pathway such that multiple pathways appear up-regulated when cells are pooled prior to bulk analyses? The scRNA-seq data we presented support the former scenario and suggest that the adaptive response to a molecular targeted therapy is more complicated than a switch to a single “bypass” growth signaling pathway. Although the co-activation of two alternate signaling pathways has been proposed, the overwhelming majority of studies focus solely on a secondary pathway driving drug tolerance. The full complement of changes that accompany the emergence of drug tolerance seems largely underestimated, which may explain why dual-targeting strategies have not made a significant clinical impact.

This activation of multiple alternate growth signaling pathways is likely an important cellular response to the sudden and sustained crash in mTOR signaling that we observed in drug-treated cells. Cell division in targeted therapy was once thought to be a simple case of incomplete inhibition or the re-activation of MAPK growth signaling. However, our proposal of alternate signaling pathways becoming activated is in agreement with more recent studies showing that MAPK-reactivation does not occur in BRAF(V600E) melanoma cells that cycle slowly in vemurafenib. We speculate that drug-tolerant cells compensate for a blockade in mTOR signaling by activating alternate growth signaling pathways to continue cell cycle signaling despite the presence of targeted therapy. 

In our study, we also observed activation of several cellular mechanisms that promote survival and are associated with slow growth, such as autophagy and senescence. Although we also observed chloroquine’s ability to enhance the gefitinib-killing of PC9 cells, our CRISPR-mediated disruption of ATG5–ATG12 association casts some doubts on the actual mechanism of increased cell death. We were initially surprised to observe a greater number of drug-tolerant ATG5-modified cells; however, functional genetic studies performed in BRAF(V600E) melanoma cell lines also showed that increased resistance—rather than increased cell death—occurred when mechanisms promoting autophagy were disrupted [33]. Conflicting results obtained with pharmacological agents on the one hand and genetic disruption methods on the other suggest that off-target pharmacological effects are likely to be involved in some strategies proposed to block drug tolerance. 

Autophagy and senescence have overlapping regulatory mechanisms and have been proposed to function together to promote chemoresistance by modulating tumor dormancy. We observed features of cellular senescence in NCSLC cells following EGFR inhibition and melanoma cells following BRAF inhibition. Additionally, we found benefits of co-treating gefitinib-tolerant PC9 cells with “senolytic” compounds to increase cell killing. However, our genetic studies against the pharmacological target casts doubt on whether these drugs are acting through their intended mechanisms. Although BCL-XL inhibitors have been used to enhance the efficacy of EGFR TKIs, our BCL-XL CRISPR knockouts survive at a higher rate, suggesting that off-target drug effects may be a concern worthy of future investigation. Moreover, it should be noted that although we observed fewer drug-tolerant cells when using the YAP inhibitor CA3 and a YAP1-knockout cell line, the greater killing efficiency observed with CA3 may indicate that some fraction of the effects is due to off-target mechanisms. In studies that use CRISPR knockout cell lines to validate drug targets, it is still worth performing side-by-side comparisons to quantify how much of the drug-killing is due to the disruption of the target versus off-target effects. Taken together, we propose that these observations should serve as a cautionary tale for studies using pharmacological inhibitors without complete genetic validation studies. 

A major concern of dual-treatment strategies is the buildup of toxic ROS. This is particularly relevant due to the emerging role that ROS-mediated cell death signaling likely has in drug-tolerant cells [12]. We and other groups have observed the ability of the ROS scavenger N-acetylcysteine to not only enhance drug tolerance but also to blunt the effects of co-treatment strategies designed to target drug-tolerant persisters [12]. These findings further highlight the importance of supporting any proposed co-treatment strategies with appropriate genetic validation studies to ensure that any observed increase in persister cell death is not simply due to drug combinations resulting in lethal ROS toxicity. 

Our work demonstrates that drug resistance to targeted therapy can involve the alteration of multiple pathways, not just one. Importantly, similar conclusions have been drawn for cytotoxic drugs as well [34]. 

To summarize, although we identified the involvement of PLC/PKC signaling in the ability of drug-tolerant cells to continue slow proliferation, our finding of numerous alternate mitogenic growth signaling pathways becoming activated simultaneously has made us hesitant to propose a treatment strategy based upon PLC/PKC signaling. 

## 4. Conclusions

Our investigation introduces a novel framework for comprehending cellular adaptations to targeted therapy through the activation of numerous alternate mitogenic pathways. We suggest that forthcoming strategies intending to counteract the drug-tolerant phenotype should employ mechanisms surpassing the targeting of a solitary growth signaling pathway. It is only through this approach that we may achieve enduring, prolonged responses in cancer patients undergoing targeted therapies.

## 5. Methods

### 5.1. Cell Culture

PC9 cells (obtained from Sigma, St. Louis, MO, USA) were cultured in RPMI (Thermo, Waltham, MA, USA) supplemented with 10% FBS (*v*/*v*) (Gemini, Dublin, Ireland) and 1% penicillin–streptomycin (Thermo). SKMEL28 cells (obtained from ATCC, Manassas, VA, USA) were cultured in MEM (Thermo) supplemented with 10% FBS (*v*/*v*) and 1% penicillin–streptomycin. Drug-naive, early-passage PC9 and SKMEL28 cells were clonally expanded and cryopreserved for drug tolerance studies. Cell lines were maintained at 37 °C in 5% CO_2_. Media changes were performed every 2–3 days. 

To generate early drug-tolerant cells, PC9 and/or SKMEL28 cells were plated on 100 mm plates (1 × 10^5^ cells/plate) and allowed to attach for 16 h. Media were replaced with fresh growth media containing gefitinib (Selleck Chemicals, Houston, TX, USA) (5 μM) or dabrafenib (Selleck Chemicals) (5 μM) for PC9 or SKMEL28 cells, respectively. Cells were cultured for the indicated amount of time, and media changes were performed every 2–3 days. 

### 5.2. Drug Co-Treatment Studies

For chloroquine co-treatment studies, PC9 cells were plated in 96-well plates at 5 × 10^4^ cells/well in regular growth media (RPMI supplemented with 10% FBS (*v*/*v*) and 1% penicillin–streptomycin) and allowed to attach for 16 h. Media were replaced with regular growth media with or without gefitinib (5 μM), containing either chloroquine (Neta Scientific, Hainesport, NJ, USA) (25 μM) or DMSO. Cells were treated for 6 days (with media changes performed every 3 days) and then quantified via CyQUANT Direct Cell Proliferation Assay (Thermo). 

For senolytic co-treatment studies, PC9 cells were first plated in 96-well plates at 1 × 10^4^ cells/well in regular growth media (RPMI supplemented with 10% FBS (*v*/*v*) and 1% penicillin–streptomycin) and allowed to attach for 16 h. Media were replaced with regular growth media with or without gefitinib (5 μM) containing either ABT263 (12.5 μM), A1331852 (12.5 μM), or DMSO as a control. Cells were grown for 3 days and then quantified via CyQUANT Direct Cell Proliferation Assay (Thermo). 

### 5.3. RNA Preparation and Bulk RNA-Seq

PC9 cells were treated with gefitinib (5 μM) and SKMEL28 cells with dabrafenib (5 μM) for 3 and 9 days (with media changes performed every 3 days). Cells were cultured in regular media for 3 days as untreated controls. Cell plating was staggered such that RNA from 3-day-treated, 9-day-treated, and untreated control samples was extracted from cells simultaneously using RNeasy Mini Kit (QIAGEN, Hilden, Germany). RNA concentration was measured with a Qubit 4 Fluorometer and associated reagents (Thermo). Library preparation was performed using the KAPA mRNA HyperPrep Kit (Roche, Basel, Switzerland). An Agilent Bioanalyzer was used to verify the RNA concentration and assess RNA sample quality. Samples were multiplexed, and sequencing was performed with 76 bp single-end reads to an estimated depth of approximately 10 million reads using the Illumina NextSeq500 (Santa Clara, CA, USA). A total of 18 samples were sequenced, three time points in triplicate, across two cell lines. 

### 5.4. Bulk RNA-Seq Data Analysis and Visualization

FASTQ files for each cell line were individually quantified using the Salmon pipeline (version 1.2.1) according to the developer’s instructions. The most up-to-date reference transcriptome assembly at the time of analysis (GRCh38.p13, release 33) was obtained directly from Gencode. Salmon quantification files were imported directly into RStudio (version 1.1.456) running R (version 3.6.1) using the *tximport* package (version 1.14.2). 

Differential gene expression and transcript abundance were determined using the DESeq2 pipeline (version 1.26.0). Significant gene expression was determined via DESeq2 Wald statistic with a padj < 0.05. Heatmaps were generated with the *ComplexHeatmap* package (version 2.2.0) using r-log (regularized log) transformed DESeq2 data. Bar plots of the mean normalized transcript abundance, with error bars representing the standard error of the mean, were generated using the *ggpubr* package (version 0.3.0). 

Gene set enrichment analysis was performed in R using the *fgsea* package (version 1.12.0). Genes with a padj < 0.05 were ranked based on Wald statistic, and 1000 permutations were performed for each analysis. GSEA plots were made using the *clusterProfiler* package (version 3.14.3). 

### 5.5. Single-Cell RNA-Seq

PC9 cells were treated with gefitinib and SKMEL28 cells with dabrafenib for 24, 48, and 72 h, along with regular media for 24 h as an untreated control. Plating and dosing were staggered such that sample collection from each time point could be performed on the same day. Upon collection, cell media were removed, and cells were briefly washed with DPBS (Gibco, Grand Island, NY, USA). Cells were incubated at 37 °C for 4 min in 1 mL of TrypLE (Gibco) to fully detach. A total of 5 mL of growth media was used to neutralize the TrypLE, and cells were spun at 300× *g* for 5 min. Cell pellets were resuspended in 1 mL cold DPBS, and cell counting was performed using a Nexelcom Cellometer (Lawrence, MA, USA). 

Single-cell suspensions were simultaneously subjected to 3′ single-cell RNA-sequencing using the 10× Chromium platform (10× Genomics). According to the manufacturer’s recommended protocols for the Single-Cell 3′ Reagent Kit v2 (10× Genomics), cell samples were loaded into the Single-Cell A chip (10× Genomics) with a recovery target of ~3000 cells. Single-Cell 3′ gel beads and emulsion oil (both from 10× Genomics) were added to the Single-Cell A chip, and a single reaction (for each cell line time course) was run on the 10× Controller to generate single-cell gel beads in emulsions (GEMS). Once GEMs were generated, reverse transcription reactions generated individually barcoded cDNA. cDNA cleanup was performed with Silane DynaBeads (Thermo Fisher) and then subjected to 12 amplification cycles according to 10× Genomics protocol recommendations. Post-cDNA amplification reaction cleanup was performed with SPRIselect beads (Beckman Coulter, Brea, CA, USA). Quantification of post-amplification cDNA was performed with a Bioanalyzer high-sensitivity chip. Pooled libraries were then sequenced on the Illumina Hi-Seq 4000 platform using paired-end sequencing with dual indexing, as recommended by the 10× Genomics protocol. 

### 5.6. scRNA-Seq Data Pre-Processing

The CellRanger 3.0.1 analysis pipeline was used to process Chromium single-cell RNA-seq output to align reads and generate feature-barcode matrices. Feature-barcode matrices for each time point were imported individually and merged using the R package “Seurat” (version 3.0). Each cell line and targeted therapy combination was processed individually. Cells with low features (less than 750 UMI counts), potential doublets (cells containing greater than 6000 UMI counts), and dead/dying cells (over 20% mitochondrial-derived UMI counts) were filtered out prior to analysis. Following these quality-control processes, a total of 10,403 single PC9 cells and 10,891 single SKMEL28 cells were used in downstream analyses. Normalization and variance stabilization were performed using SCT transformation.

### 5.7. scRNA-Seq Analysis

Several dimensionality reduction methods were used to explore our single-cell RNA-seq. Principal component analysis was performed using the built-in “RunPCA” function in the Seurat package. Scatter plots of the PCA analysis were generated using the “DimPlot” command, specifying “pca” as the reduction method. Similarly, tSNE analysis was performed using the built-in “RunTSNE” function of the Seurat package. tSNE analysis was performed using principal components 1–13, which were deemed significant when evaluating elbow plots of the top 40 principal components. Finally, UMAP reduction was performed using the python-based “umap-learn” package. This was performed in R using the “reticulate” package and the “RunUMAP” command built into Seurat version 3. The number of neighbors was set to 15 (for PC9 cells) and 50 (for SKMEL28 cells). The number of epochs used was 500, and the minimum distance was set to 0.3 for each cell line. The first 13 principal components were used. The generation of clusters and identification of top cluster markers were performed using the built-in functions of the Seurat package. 

Cluster marker genes were identified using the Seurat command “FindAllMarkers” with a logfc threshold of 0.25 (including only genes with an increase in transcript count) and an adjusted *p*-value of <0.01. 

Single-cell RNA-seq gene–gene relationships were visualized using the python-based tool, MAGIC. scRNA-seq counts were imported, normalized, and transformed using the default parameters. MAGIC was run on the data using the default values (knn = 5; knn_max = 3 × knn; decay = 1; t = 3). Gene–gene relationships were plotted, with each point colored by time in the drug.

### 5.8. Reverse Phase Protein Array Analysis

PC9 cell plating and drug time courses were staggered such that cells were pelleted simultaneously prior to RPPA analysis. For each sample, three 100 mm plates were pooled prior to pelleting to ensure enough bulk material was collected. Cell pellets were immediately stored at −80 °C and shipped on dry ice to the RPPA Core Facility at MD Anderson Cancer Center. Protein extraction, reverse phase protein array, and quality control was performed on a total of 12 samples (untreated = 3; 3-day gefitinib = 3; 6-day gefitinib = 3; and 9-day gefitinib = 3) by the RPPA Core Facility. Heatmaps of normalized log2 values were generated with the *complexHeatmaps* package in R. The scatter plots of RPPA and RNA data were generated using the differences in the mean of normalized linear log2-centered values between 9-day gefitinib-treated and untreated PC9 cells. Log-fold change values for RNA transcripts were generated using DESeq2. Boxplots of protein levels were generated with *ggpubr*, using the mean log2-centered values for each time point.

### 5.9. Antibodies and Immunoblotting

The following antibodies were used for immunoblotting: Phospho-S6 Ribosomal Protein Ser235/236 Rabbit mAb (CST, #4858), Phospho-S6 Ribosomal Protein Ser240/244 Rabbit mAb (CST, #5364), BCL-xL (CST #2764), ATG5 (CST #12994) Beta-Actin Rabbit mAb (CST, #4970), and Anti-rabbit IgG HRP-linked Antibody (CST #7074). The antibodies were diluted 1:1000 for immunofluorescence and immunoblotting. 

Cells were detached by briefly washing with DPBS and incubating at 37 °C for 4 min in 1 mL of TrypLE. A total of 5 mL of growth media was used to neutralize the TrypLE, and cells were spun at 300× *g* for 5 min. Cells pellets were placed on ice and lysed with RIPA buffer (Thermo) containing Halt protease and phosphatase inhibitor (Thermo). Cell lysate was quantified using Bradford assay (Bio-Rad, Hercules, CA, USA) prior to the addition of Bolt Reducing Agent (500 mM DTT) and LDS (Thermo). Samples were separated on 4–12% Bolt Bis-Tris Plus Gels (Thermo) and subsequently transferred to a nitrocellulose membrane using the iBlot2 transfer module (Thermo). The blot was washed with TBST (Bio-Rad) and then blocked with 5% *w*/*v* BSA (Sigma) for 60 min at RT. After a brief wash with TBST, blots were incubated with primary antibody (1:1000) overnight at 4 °C. The following day, blots were washed with TBST 4 times and then incubated with HRP-linked secondary antibody (1:5000) at RT for 1–2 h. Following another 4 washes with TBST, immunoreactive bands were detected using the Clarity ECL substrate (Bio-Rad). 

Appendix A contain the original, uncropped images of immunoblots. 

### 5.10. CRISPR Studies

gRNA sequences were designed and ordered as custom DNA oligos (Thermo). gRNA components were annealed and cloned into the lentiCRISPRv2 plasmid (Addgene ##52961, Watertown, MA, USA), according to the instructions from the Zhang Lab. lentiCRISPRv2, along with helper plasmids VSVG (Addgene #8454) and psPAX2 (Addgene #12260), were transfected into 293T cells using the ProFection Mammalian Transfection System (Promega, Madison, WI, USA). Packaged lentivirus was harvested and transduced into PC9 cells (5 × 10^4^ cells/well in 6 well plates). Following 9-day selection in regular growth media supplemented with puromycin (5 μg/mL), surviving cells were plated in 96-well plates (at 0.5 cells/well) to generate clonal colonies. Single-cell colonies were verified via inspection with a microscope and the cells were expanded, with media being changed every 3 days. Expanded colonies were passaged from 96-well to 24-well plates and eventually expanded in 100 mm dishes prior to cryopreservation and use in future studies. 

ATG5 and BCL-XL CRISPR-modified cell lines, as well as control PC9 cells, were plated in 96-well plates at 5 × 10^4^ cells/well in regular growth media and allowed to attach for 16 h. Media were replaced with growth media with or without gefitinib (5 μM). Cells were grown for 3 days and then quantified via CyQUANT™ Direct Cell Proliferation Assay (Thermo) to assess differences in cell numbers between control and CRISPR-modified PC9 cell lines.

### 5.11. Visualizing Autophagy and Senescence

To visualize autophagy or senescence in cells developing drug tolerance, PC9 cells were plated in 6-well plates at 1.15 × 10^4^ cells/well in regular growth media and allowed to attach for 16 h. Media were replaced with regular growth media with or without gefitinib (5 μM). Media were replenished every 2–3 days. At the indicated time in drug, cells were stained to visualize autophagy or senescence. Autophagic flux was visualized using the CYTO-ID Autophagy Detection Kit (ENZO, #ENZ-51031-0050). Staining was performed according to the manufacturer’s instructions. Representative brightfield, DAPI, and GFP images were captured for each well, and channels were merged using ImageJ (https://imagej.net/ij/, accessed on 20 February 2024).

Senescence was visualized using the commercially available Cellular Senescence Activity Assay (Enzo, # ENZ-KIT129-0120) at each indicated time point according to the manufacturer’s instructions. Representative brightfield images were captured for each well.

To visualize the effects on autophagy of the ATG5 CRISPR-modification, relative to parental PC9 cells, each cell line was plated at 7.5 × 10^5^ cells/well in 6-well plates in regular media and allowed to attach for 16 h. Negative control cells were grown in regular growth media and positive control cells were grown starvation media consisting of normal growth media supplemented with rapamycin (500 nM) and chloroquine (10 μM) according to the manufacturer’s instructions. Regular growth media were replaced with starvation media 16 h prior to imaging to induce autophagic signal. Autophagy was visualized using the CYTO-ID Autophagy Detection Kit (ENZO, ENZ-51031-0050). Staining was performed according to the manufacturer’s instructions. Representative brightfield, DAPI, and GFP images were captured for each well, and channels were merged using ImageJ.

## Figures and Tables

**Figure 1 cancers-16-01001-f001:**
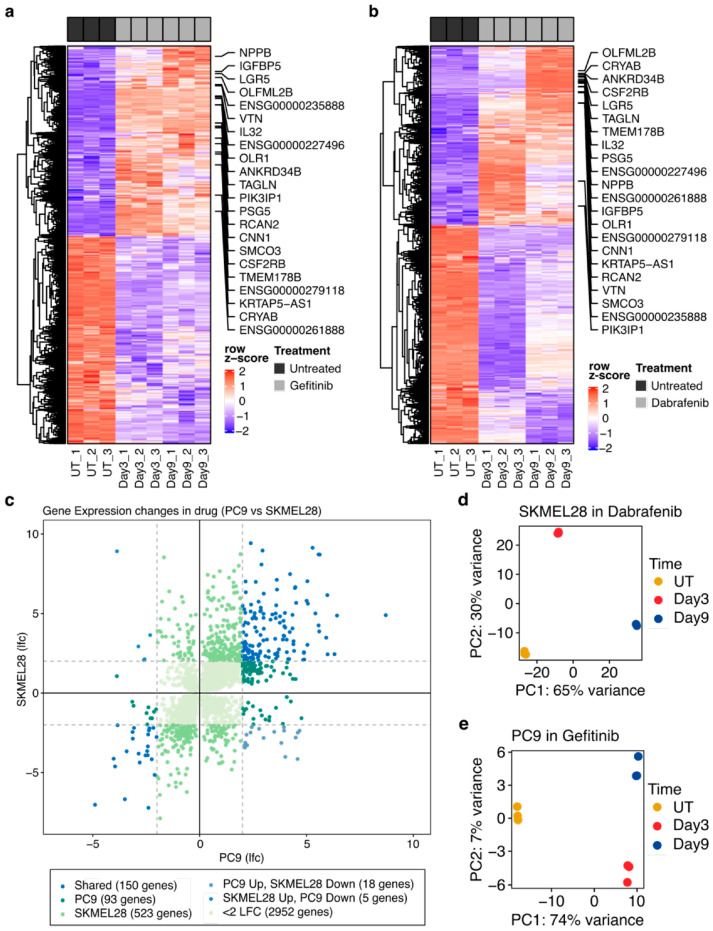
Shared transcriptional alterations between PC9 and SKMEL28 cells during the development of drug tolerance. (**a**) Heatmap of the commonly altered genes in PC9 cells either untreated (UT) or treated with gefitinib after 3 days or 9 days. (**b**) Heatmap of commonly altered genes in SKMEL28 cells either untreated (UT) or treated with dabrafenib after 3 days or 9 days. (**c**) Scatter plot showing the difference between PC9 cells treated with gefitinib and SKMEL28 cells treated with dabrafenib with respect to the log-fold changes in the expression of commonly altered genes. (**d**) PCA plot of SKMEL28 cells with each point representing a unique sample colored by time spent in dabrafenib. The first principal component (PC1) is associated with 65% of the variance and separates samples based on the presence or absence of drug treatment (UT = untreated control). (**e**) PCA plot of PC9 cell samples, with each point representing a unique sample colored by time spent in gefitinib (UT = untreated control). The first principal component (PC1) is associated with 74% of the variance and separates samples based on the presence or absence of drug treatment.

**Figure 2 cancers-16-01001-f002:**
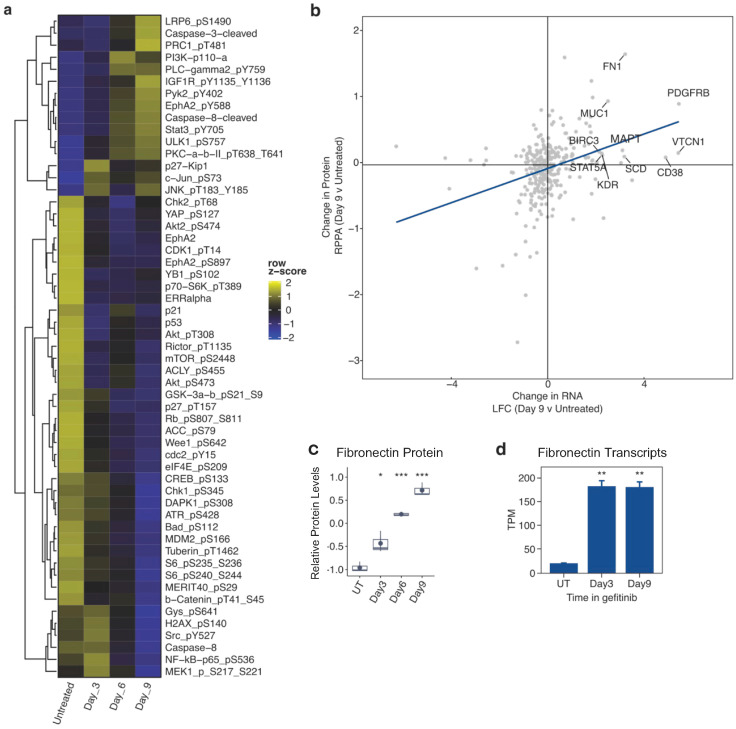
Proteomic alterations during the development of gefitinib tolerance in PC9 cells. (**a**) Heatmap of hierarchical clustering of all significantly altered (*p*-value < 0.05) phosphoproteins assayed by RPPA. Columns represent the average (n = 3) for each time point, and row colors correspond to the z-score. (**b**) Scatter plot comparing the log-fold change values of mRNA (day 9 versus untreated) to the changes in protein levels as quantified by RPPA. Several members that are up-regulated in both datasets are labeled. (**c**) RPPA quantification of fibronectin protein in PC9 cells at the indicated times of treatment with gefitinib (UT = untreated). (**d**) RNA abundance levels (transcripts per million reads) of the fibronectin gene (FN1) in PC9 cells either untreated (UT) or treated with gefitinib for 39 days. Data are represented as the mean ± SEM (n = 3). A two-tailed Student’s *t* test was used for comparisons between the two groups. * *p* < 0.05, ** *p* < 0.01, *** *p* < 0.001.

**Figure 3 cancers-16-01001-f003:**
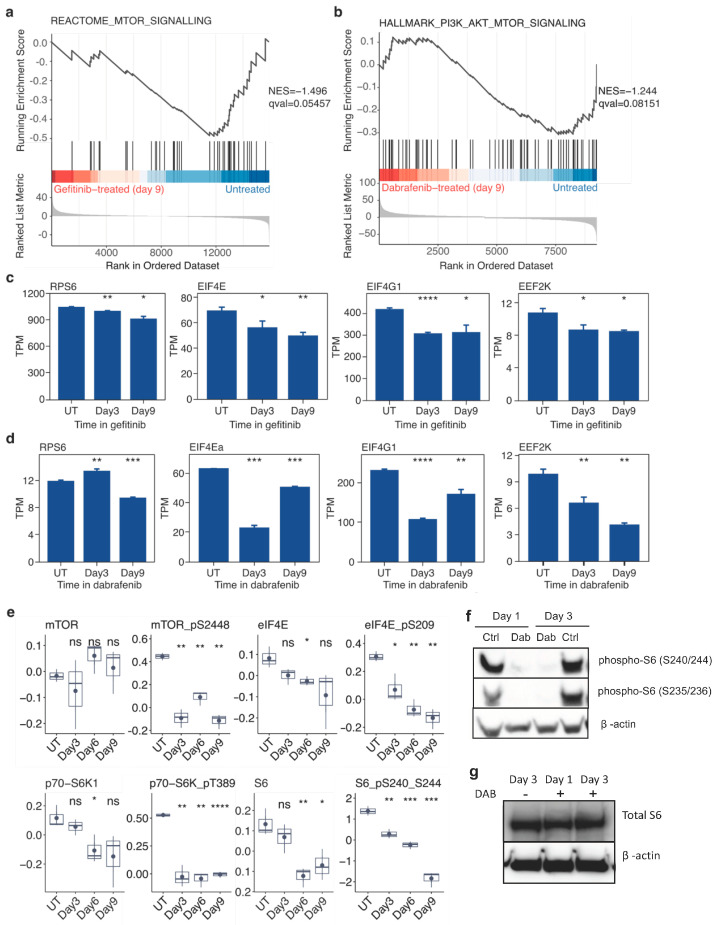
Gefitinib in PC9 cells and dabrafenib in SKMEL28 cells induce sustained mTOR suppression during emergence of drug-tolerant cells. (**a**) GSEA NES scores indicate that the Reactome gene set “mTOR signaling” is enriched in untreated PC9 cells when compared to PC9 cells treated with gefitinib for 9 days. (**b**) GSEA NES scores indicate that the hallmark gene set “PI3K/AKT/mTOR signaling” is enriched in untreated SKMEL28 cells when compared to SKMEL28 cells treated with dabrafenib for 9 days. (**c**,**d**) Normalized transcript abundance levels of several genes associated with mTOR signaling, including RPS6, EIF4E, EIF4EG1, and EEF2K, in PC9 cells in gefitinib (**c**) or SKMEL28 cells in dabrafenib (**d**) for 3 or 9 days compared to untreated controls. (**e**) RPPA quantification of proteins and phosphoproteins associated with mTOR signaling, including mTOR, phospho-mTOR (S2848), eIF4E, eIF4E (S209), p70-S6K1, phospho-p70-S6K (T389), S6, and phospho-S6 (S240/S244) from PC9 cells in gefitinib for 0, 3, 6, or 9 days. (**f**) Western blot analysis of phospho-S6 (S235/236) and phospho-S6 (S240/244) from untreated SKMEL28 (UT) and SKMEL38 cells in dabrafenib (D) for 24 h (top) or 72 h (bottom). (**g**) Western blot analysis of total S6 protein from untreated SKMEL28 and SKMEL38 cells in dabrafenib for 24 h or 72 h (see Appendix A for unmodified original Western blot). Data are represented as the mean ± SEM (n = 3). A two-tailed Student’s *t* test was used for comparisons between the two groups. * *p* < 0.05, ** *p* < 0.01, *** *p* < 0.001, **** *p* < 0.0001; ns = *p* > 0.05.

**Figure 4 cancers-16-01001-f004:**
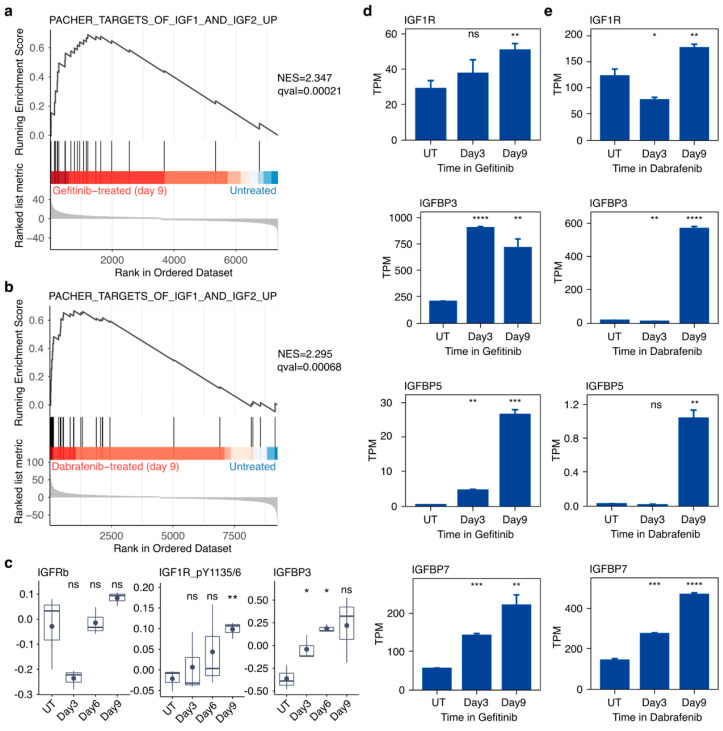
Gefitinib in PC9 cells and dabrafenib in SKMEL28 cells induces activation of the IGF signaling pathway in drug-tolerant cells. (**a**) GSEA NES scores indicate that the gene set “Targets of IGF1 and IGF2 Up” is significantly enriched in PC9 cells treated with gefitinib for 9 days relative to untreated PC9 cells. (**b**) GSEA NES scores indicate that the gene set “Targets of IGF1 and IGF2 Up” is significantly enriched in SKMEL28 cells treated with dabrafenib for 9 days relative to untreated SKMEL28 cells. (**c**) RPPA quantification of proteins and phosphoproteins associated with IGF signaling, including IGFRb, phospho-IGF1R (Y1135/6), and IGFBP3, from PC9 cells in gefitinib for 3, 6, or 9 days along with untreated control cells (UT). (**d**,**e**) Normalized transcript abundance levels of several genes associated with IGF signaling, including IGF1R, IGFBP3, IGFBP5, and IGFBP7, in PC9 cells in gefitinib (**d**) or SKMEL28 cells in dabrafenib (**e**) for 3 or 9 days compared to untreated control cells (UT). Data are represented as the mean ± SEM (n = 3). A two-tailed Student’s *t* test was used for comparisons between the two groups. * *p* < 0.05, ** *p* < 0.01, *** *p* < 0.001, **** *p* < 0.0001; ns = *p* > 0.05.

**Figure 5 cancers-16-01001-f005:**
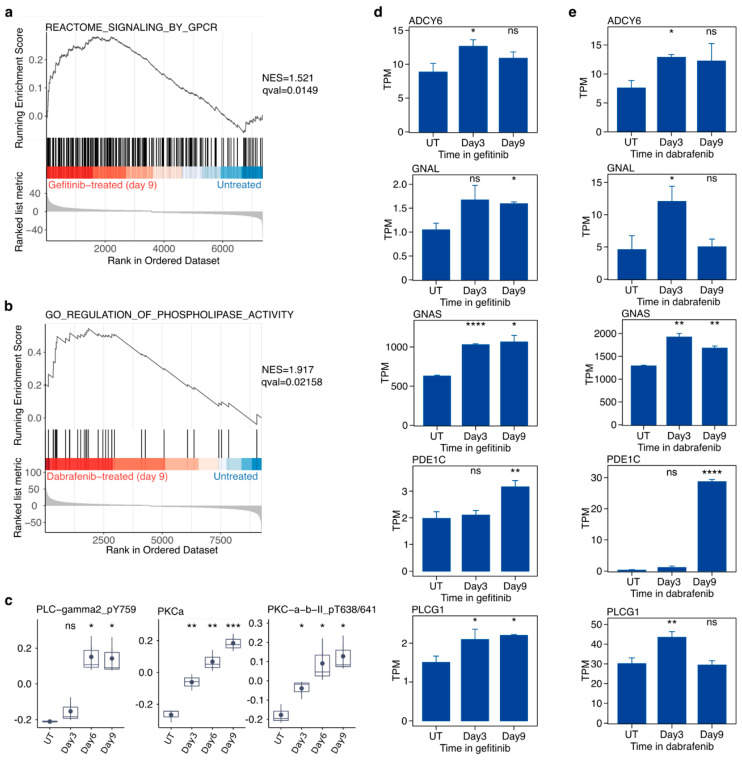
Gefitinib in PC9 cells and dabrafenib in SKMEL28 cells induces activation of PLC/PKC signaling in drug-tolerant cells. (**a**) GSEA NES scores indicate that the Reactome gene set “Signaling by GPCR” is significantly enriched in PC9 cells treated with gefitinib for 9 days relative to untreated PC9 cells. (**b**) GSEA NES scores indicate that the GO gene set “Regulation of Phospholipase Activity” is significantly enriched in SKMEL28 cells treated with dabrafenib for 9 days relative to untreated SKMEL28 cells. (**c**) RPPA quantification of proteins and phosphoproteins associated with PLC/PKC signaling, including phospho-PLC-gamma2 (Y759), PKCa, and PKCa/b (T638/641), from PC9 cells in gefitinib for 3, 6, or 9 days along with untreated control cells (UT). (**d**,**e**) Normalized transcript abundance levels of several genes associated with PLC/PKC activity including ADCY6, GNAL, GNAS, PDE1C, and PLCG1 in PC9 cells in gefitinib (**d**) or SKMEL28 cells in dabrafenib (**e**) for 3 or 9 days along with untreated control cells (UT). Data are represented as the mean ± SEM (n = 3). A two-tailed Student’s *t* test was used for comparisons between the two groups. * *p* < 0.05, ** *p* < 0.01, *** *p* < 0.001, **** *p* < 0.0001; ns = *p* > 0.05.

**Figure 6 cancers-16-01001-f006:**
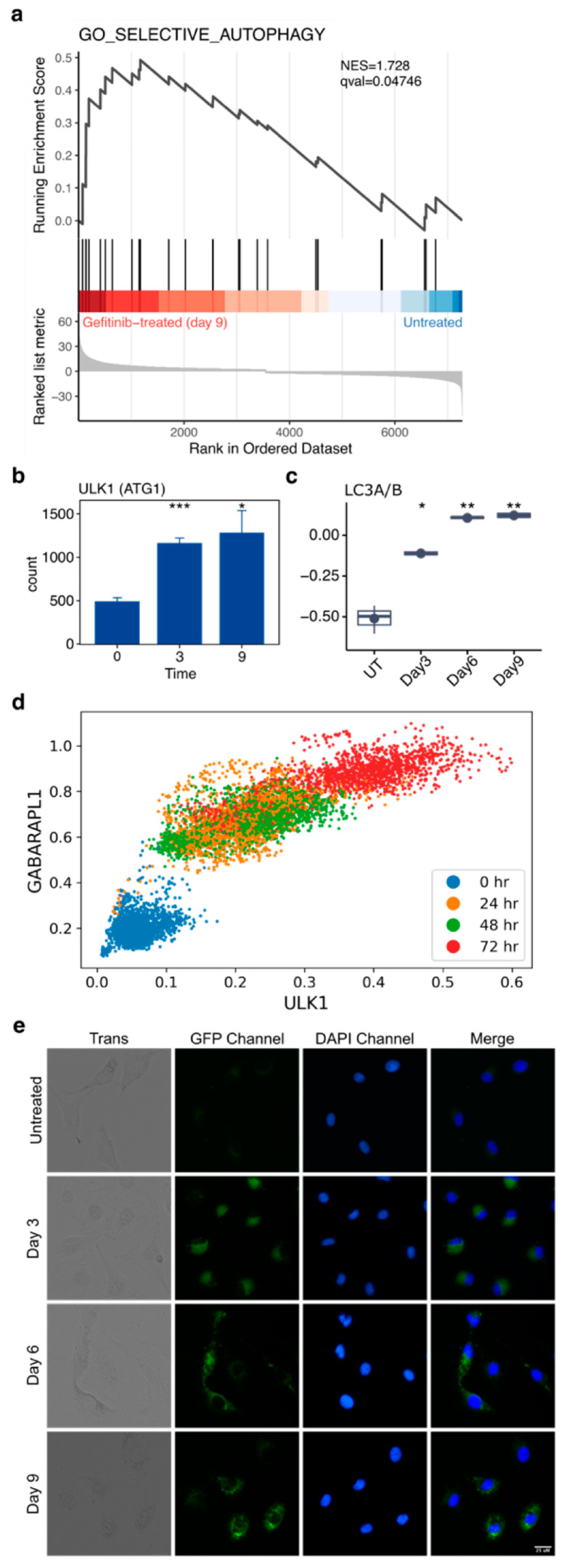
Increased autophagic flux accompanies the emergence of drug tolerance in PC9 cells treated with gefitinib. (**a**) GSEA plot reveals that the GO gene set “Selective Autophagy” is significantly enriched in PC9 cells treated with gefitinib for 9 days relative to untreated PC9 cells. (**b**) Normalized transcript abundance of ULK1 in PC9 cells treated with gefitinib for 0, 3, or 9 days. (**c**) RPPA quantification of LC3A/B from PC9 cells in gefitinib for 0, 3, 6, or 9 days. (**d**) Gene–gene relationship of ULK1 and GABARAPL1 from MAGIC imputation analysis performed on scRNA-seq of PC9 cells in gefitinib for 0, 24, 48, or 72 h. (**e**) Autophagy flux in PC9 cells in gefitinib for 0, 3, 6, or 9 days visualized via CYTO-ID staining. Data are represented as the mean ± SEM (n = 3). A two-tailed Student’s *t* test was used for comparisons between the two groups. * *p* < 0.05, ** *p* < 0.01, *** *p* < 0.001; ns = *p* > 0.05.

**Figure 7 cancers-16-01001-f007:**
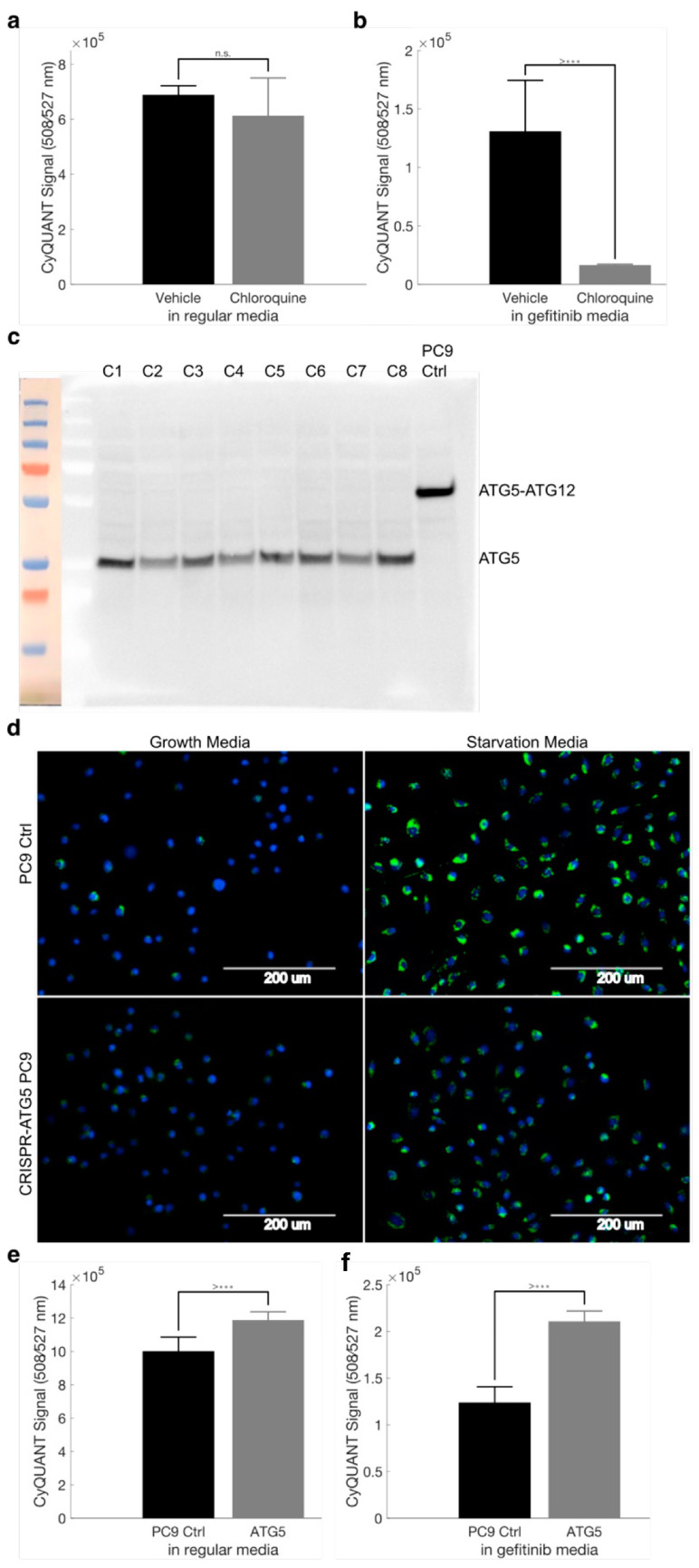
Pharmacologic but not genetic disruption of autophagy results in increased gefitinib efficacy. (**a**) Quantification of PC9 cell growth in normal media treated with DMSO or 25 μM chloroquine. (**b**) Quantification of PC9 cell growth in gefitinib media treated with DMSO or 25 μM chloroquine. Data are represented as the mean ± SD (n = 10). A two-tailed Student’s *t* test was used for comparisons between the two groups. *** *p* < 0.001; n.s. = *p* > 0.05. (**c**) Western blot analysis of ATG5 protein from parental PC9 cells or clonal populations (C1-C8) of CRISPR-ATG5 PC9 cells. (**d**) Autophagic flux in parental PC9 cells and CRISPR-ATG5 PC9 cells in normal growth media and autophagy-inducing starvation media, visualized with the CYTO-ID autophagy detection kit. (**e**,**f**) Quantification of PC9 cell growth and CRISPR-ATG5 PC9 cell growth (Clone 2) in normal media. Data are represented as the mean ± SD (n = 8). A two-tailed Student’s *t* test was used for comparisons between the two groups. *** *p* < 0.001; n.s. = *p* > 0.05. The uncropped bolts are shown in Appendix A.

**Figure 8 cancers-16-01001-f008:**
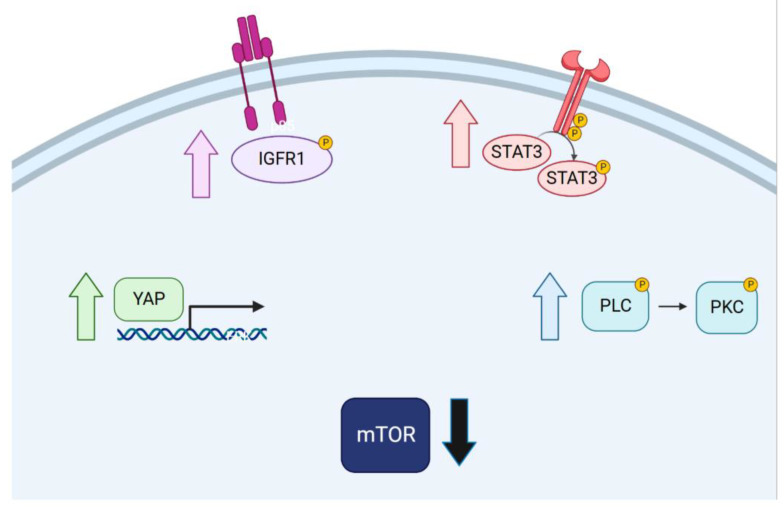
Graphical summary of the compensatory pathways activated upon treatment. Four pathways are activated during the development of drug tolerance in both PC9 lung cancer cells and SKMEL28 melanoma cells: IGFR1, YAP, STAT3, and PLC/PKC. The mTOR pathway is down-regulated in drug-tolerant cells.

**Table 1 cancers-16-01001-t001:** Changes in gene expression of select genes in both lung and melanoma persisters.

	Lung		Melanoma	
Gene	*p*-Val	logFC	*p*-Val	logFC
APOBEC3A	0.38	1.47	0.13	0.59
APOBEC3B	0.02	2.16	2 × 10^−3^	0.38
AXL	0.01	0.65	9 × 10^−5^	15.98
GAS6	2 × 10^−3^	2.36	9 × 10^−5^	4.93
GPX4	0.06	1.30	0.09	1.28
GSR	0.08	0.79	0.02	1.43
GSS	0.94	1.01	0.02	0.61
GSTA4	0.00	2.51	0.00	3.60
GSTCD	0.01	0.67	0.01	0.75
GSTK1	0.01	2.03	0.68	0.95
GSTM1	1 × 10^−3^	4.93	0.25	0.63
GSTM2	9 × 10^−5^	4.86	0.09	1.27
GSTM3	0.01	2.67	0.01	1.63
GSTM4	1 × 10^−3^	7.18	0.02	0.65
GSTO1	0.17	0.83	1 × 10^−3^	0.19
GSTO2	0.14	0.65	0.04	2.16
GSTP1	0.13	1.22	0.69	0.95
GSTZ1	0.18	1.25	0.64	1.09
MGST1	0.34	0.89	0.33	0.89
MGST2	0.82	1.04	0.00	0.23
MGST3	0.04	1.73	0.29	0.86
POLE	0.05	0.72	2 × 10^−3^	0.27
POLE2	0.01	0.56	4 × 10^−3^	0.34
POLE3	4 × 10^−3^	0.62	0.11	1.24
POLE4	0.32	0.83	0.01	0.61
POLH	0.17	0.81	0.17	0.82
POLI	0.72	1.04	8 × 10^−4^	2.10
POLK	0.18	0.83	0.01	1.57
POLQ	5 × 10^−3^	0.54	2 × 10^−3^	0.35
RAD18	0.04	0.71	0.02	1.45
REV1	0.32	0.86	0.44	1.06

## Data Availability

Data are contained within the article. Raw FASTQ files for bulk and single-cell RNA-seq studies, along with processed data files, are available for download from the Gene Expression Omnibus (GSE162045).

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
