# Peer review of "Induction of Multiple Alternative Mitogenic Signaling Pathways Accompanies the Emergence of Drug-Tolerant Cancer Cells"

_cancers, 2024, doi:10.3390/cancers16051001_

Round 1

Reviewer 1 Report

Comments and Suggestions for Authors

Drug resistance can emerge from a subset of cancer cells initially surviving drug treatment and subsequently developing into a pool of drug-tolerant cells. Recent studies have identified activation of specific bypass pathways as critical therapeutic targets for preventing drug tolerance. Taking a systems biology approach, authors have utilized proteomics and genomics to investigate the development of drug tolerance to EGFR inhibitors in EGFR-mutant lung adenocarcinoma cells and BRAF inhibitors in BRAF-mutant melanoma cells. Their findings reveal the activation of numerous alternative mitogenic pathways, including YAP, STAT3, IGFR1, and phospholipase C (PLC)/protein kinase C (PKC) pathways, in both cases. These results suggest that an effective therapeutic strategy to prevent drug tolerance cells. However, further analysis is required to provide concrete evidence of the involvement of these pathways in cancer resistance.

Major Comments: 

1. Running an animal experiment to validate the findings specifically with the knock-out clones to validate the findings would significantly improve the manuscript. 

2.  Studies have reported role of metabolism and AXL and persister cells. Have authors specifically looked for glutathione markers and AXL in their data. Can the link be established to persister cells ? 

3. Peristers cells are related to APOBEC3A, 3B, RAD18 and Low fidelity polymerasis. Authors should include these molecules in the analysis.  

4. Persister and IGF1R is very well known. Authors have to move this to the supplementary and focus more on establishing the link between persisters and autophagy and include in main figure.

Minor Comments: 

1. Several contributions related to lung cancer and persisters are missing and has to cited.  

2. High quality images are preferable.  

Reviewer 2 Report

Comments and Suggestions for Authors

In this manuscript, the authors explore how drug resistance can develop in cancer cells. Using proteomics and genomics, the authors investigated drug tolerance in lung adenocarcinoma and melanoma cells. They discovered multiple alternative pathways becoming activated, indicating that preventing drug tolerance requires targeting various pathways rather than just one. The introduction provides useful background information and a sufficient review of previous papers relevant to this topic. The objective of this study, to determine the entire scope of mitogenic signaling pathways upregulated in response to the blockade of mutant EGFR and mutant BRAF signaling, is very important and interesting. To achieve this, the authors conducted bulk RNA-sequencing and Reverse Phase Protein Array on cells surviving after drug treatment and found mTOR, PLC/PKC, STAT, and YAP signaling commonly altered. Importantly, through single-cell RNA sequencing, they concluded that the enriched pathways are simultaneously upregulated in single cells, rather than being upregulated in different sub-populations prior to pooled analysis. Using this information, the authors attempted to find effective ways to overcome drug resistance. The method is clear, and the results were well described, but there is room for revision.

1.      There are currently no figure legends. Figure legends should be inserted.

2.      In the part discussing autophagy and senescence, the effect of chemical and genetic inhibition resulted in different outcomes. I think it is important to present negative results as they are, but they should be presented in a more organized manner. I recommend combining each two subtitles into one, as the authors presented in Figure S6: Pharmacologic but not genetic inhibition of BCL-XL results in increased gefitinib efficacy.

3.      This paper demonstrates that drug resistance to targeted therapy can be acquired through multiple pathways, not just one. It would be beneficial to mention that similar conclusions have been drawn for cytotoxic drugs as well. [Reference: Exp Mol Med. 2020 Jul;52(7):1102-1115].

Round 2

Reviewer 1 Report

Comments and Suggestions for Authors

The authors have addressed the comments. However there are missing pieces to complete the manuscript.

1. Drug-tolerant cells show significant gene expression changes affecting AXL/GAS6, enzymes that utilize glutathione as a co-factor, and low-fidelity polymerases

This part has a subheading in the manuscript, but the figure or gene indication is supplementary. The authors have to indicate this in the main figure.  

2. Authors have added the missing references. However, there are few important contibustions to the field and has to be included. 

Missing references.  

https://pubmed.ncbi.nlm.nih.gov/38049664/

https://pubmed.ncbi.nlm.nih.gov/37407818/

https://pubmed.ncbi.nlm.nih.gov/37461590/

Author Response

We'd like to thank the reviewer for quickly responding to our reply to the first round of reviews.  In terms of the second round, we have addressed this reviewer's comments as follows:

Point 1. We have moved the table corresponding to "Drug-tolerant cells show significant gene expression changes affecting AXL/GAS6, enzymes that utilize glutathione as a co-factor, and low-fidelity polymerases" from supplementary information to the main text.

Point 2.  We have added the three references.

Reviewer 2 Report

Comments and Suggestions for Authors

The authors fully addressed my concerns, and now I think this manuscript is suitable for publication.

Author Response

We'd like to thank the reviewer for his efforts, they have been much appreciated.